# Uncoordinated Coupling Assessment of New Urbanization and Ecological Carrying Capacity in the Yellow River Basin

**DOI:** 10.3390/ijerph19159016

**Published:** 2022-07-25

**Authors:** Dongmin Zhang, Libo Zhu, Xiuying Ma, Zuoming Liu, Hongwei Cui

**Affiliations:** 1School of Statistics, Jilin University of Finance and Economics, Jingyue Street 3699, Changchun 130117, China; a46775298@163.com (L.Z.); mxy9017@163.com (X.M.); 2School of Business and Management, Jilin University, Qianjin Street 2699, Changchun 130012, China; lzmhero_@126.com; 3School of Finance, Jilin Business and Technology College, Kalun Lake Street 1666, Changchun 130507, China

**Keywords:** Yellow River Basin, new urbanization, ecological carrying capacity, non-coordinated coupling

## Abstract

Under the restriction of the national “double carbon” goal, how to realize the coordination between urbanization and low-carbon development in the Yellow River Basin is a problem worthy of attention. In this paper, a new urbanization and ecological carrying capacity evaluation index system is established to evaluate the new urbanization level and ecological carrying capacity of the Yellow River Basin. On this basis, the uncoordinated coupling level of new urbanization and ecological carrying capacity in the Yellow River Basin is measured by using the improved uncoordinated coupling model, and its temporal and spatial characteristics and internal impact mechanism are analyzed. The study shows that the new urbanization and ecological carrying capacity of the Yellow River Basin has a benign development trend as a whole. Shandong province belongs to the low-level uncoordinated coupling type; Gansu Province and Qinghai Province belong to the running-in uncoordinated type; and Shanxi Province, the Inner Mongolia Autonomous Region, Shaanxi Province, and the Ningxia Hui Autonomous Region belong to the antagonistic uncoordinated coupling type. The uncoordinated coupling degree between new urbanization and ecological carrying capacity in the Yellow River Basin has a spatial interaction effect. It presents a low-level cluster centered on Shaanxi Province and Shandong Province and a high-level cluster centered on Gansu Province, Qinghai Province, and the Ningxia Hui Autonomous Region. From the perspective of the internal main impact mechanism, water resources have a two-way impact on the development of the two systems of new urbanization and ecological carrying capacity; the number of permanent residents and the level of scientific and technological investment have a one-way impact on the process of new urbanization; and the green coverage rate of built-up areas has a one-way impact on the development of ecological carrying capacity. The main contributions of this paper are as follows. First, the evaluation index system of new urbanization and ecological carrying capacity has been improved in combination with the new development concept. The evaluation of new urbanization by this index system is more in line with the current national requirements for high-quality development. Second, the impact of potential resources and human regulation has been added to the traditional ecological carrying capacity evaluation index system, and the evaluation of ecological carrying capacity by this index system is more in line with reality. Thirdly, taking the time effect into account, an improved uncoordinated coupling method is proposed. Using this method to evaluate the relationship between systems is conducive to bringing the dynamic relationship within the system into the evaluation system, which is more in line with the reality of system changes. Fourth, from the perspective of problem diagnosis, research on the relationship between new urbanization and ecological carrying capacity will help to find the internal mechanism that affects the coordinated development of new urbanization and ecological carrying capacity in the Yellow River Basin. This method is universal for exploring the internal influence mechanism of the relationship between systems.

## 1. Introduction

Urbanization is a critical path for the rapid development of a country. China has entered the stage of building a moderately prosperous society in all aspects and is currently in an important stage of economic transformation, upgrading, and further promotion of socialist modernization, which is also a critical period for the in-depth development of urbanization construction. Since China released the National New Urbanization Plan 2014–2020 in 2014, new urbanization projects have become key in all regions. The traditional definition of urbanization simply refers to the process of a rural population transforming into an urban population. New-type urbanization takes urban–rural integration, industrial interaction, conservation and intensification, ecological livability, and harmonious development as its basic characteristics, and the coordinated development and mutual promotion of large, medium, and small cities, towns, and new rural communities is its goal. As the connotation of new urbanization continues to deepen, in 2020, the Fifth Plenary Session of the 19th Central Committee of the Communist Party of China proposed to promote a new type of urbanization with people at its core, build a new pattern of territorial space development and protection, and promote the coordinated development of the region. The Yellow River Basin has been an important birthplace of civilization in China since ancient times. It is also an important object of China’s policy of arranging industrialization and modernization, so the promotion and high-quality development of new urbanization in the region has been of great concern. According to a research report released by the Development Research Center of the State Council, the urbanization of the Yellow River Basin is now in the middle and late stages of rapid development. It is necessary to change the way of planning based on “population growth”; accelerate the application and promotion of ecological technology, new energy, and digital technology in the Yellow River Basin; promote the change in production and lifestyle in the whole basin; and reduce the impact of urbanization on the population. The government work report of the 13th National People’s Congress proposes to solidly promote environmental protection and high-quality development in the Yellow River Basin and to deeply promote a new urbanization strategy with people at its core. Therefore, new urbanization in the Yellow River Basin should move from the original criteria of population number growth to the direction of high-quality development, such as people-centered development and green development.

The Yellow River Basin is an important ecological barrier and economic zone in China and a major material production area for resources and food in China. The high-quality development and ecological protection of the Yellow River Basin are major national strategies, along with the coordinated development of the Beijing–Tianjin wing, the development of the Yangtze River Economic Belt, the construction of the Guangdong–Hong Kong–Macao Bay Area, and the integration of the Yangtze River Delta. The Yellow River Basin has been a disaster-prone area since ancient times, but now, under the economic construction of the country, the occurrence of disasters has been greatly reduced. However, with the new urbanization of economic construction activities, there are still ecological and environmental problems. For example, in 2019, the central inspection team found that enterprises do not follow the plan of mining in Gansu Province; there are abandoned slag piles, wastewater discharge, and other pollution problems, all of which are more serious than before. The progress of remediation work is slow, and the environmental risks are high. In this context, it is of great practical significance to study the coupling and coordination between new urbanization and ecological resources and the environment in the Yellow River Basin.

Based on this background, this paper constructs a new urbanization evaluation index system from the perspective of the human core. The single element carrying capacity combined with the total carrying capacity is used to measure the regional ecological, resource, and environmental levels. The evaluation index system of ecological carrying capacity is reconstructed to measure cities’ new urbanization and ecological carrying capacity levels in the Yellow River Basin. The uncoordinated coupling degree model is used to measure the uncoordinated coupling degree between the new urbanization and ecological carrying capacity of the Yellow River Basin. The problems of the coordinated development of the new urbanization and ecological carrying capacity of the Yellow River Basin are analyzed from the perspective of reverse thinking. Suggestions are made to promote the coordinated development of the new urbanization and ecological carrying capacity of the Yellow River Basin.

At present, there have been a large number of studies on the coupling and coordination between new urbanization and the ecological environment system. This paper summarizes two aspects: constructing an index system of new urbanization and ecological carrying capacity. Due to the different definition categories of new urbanization and ecological carrying capacity, scholars have not formed a unified and standard system for constructing indicator systems of the two systems. In terms of new urbanization, most scholars construct indicator systems around four aspects, population, economy, society, and space, but there are some differences in selecting specific indicators. For example, Zhang and Jiao [1] decomposed the urbanization system into four subsystems of population urbanization, economic urbanization, spatial urbanization, and social urbanization from the perspective of inter-provincial urbanization and selected the corresponding indicators. The conclusion was that the coupling and coordination degree of inter-provincial urbanization and the ecological environment system in China showed an increasing trend. Tan et al. [2] concluded that the spatial concentration of population and employment structure transformation in the process of population urbanization are often closely linked with urban spatial expansion, so the population and spatial urbanization process is reflected by the population urbanization index system. It was concluded that the coupling coordination degree of urbanization and ecological environment in Jilin Province showed a continuous growth trend and developed from basic incoordination to advanced coordination. Zhang et al. [3] constructed a triangular model of urbanization quality based on the efficiency-level model and divided it into two subsystem indicators of urbanization level and urbanization efficiency. They concluded that the distribution of the coupling degree of new urbanization quality and ecological environment carrying capacity in Chongqing city and counties is basically in line with the spatial differentiation rule of “one circle and two wings”. Wu [4] combined the development status of each district and city in Fujian Province and constructed the urbanization index system from five aspects, namely population, economic, infrastructure, social, innovation, and globalization factors. It was concluded that the coordinated development between urbanization and the resource and environmental carrying capacity of Fujian Province has been improving. Zhao et al. [5] drew on the existing results to construct a new urbanization level system from the perspectives of three subsystems: the urbanization level, public service level, and infrastructure level. In terms of ecological carrying capacity, Xiong et al. [6] constructed the ecological carrying capacity of the Dongting Lake area from three perspectives: the support capacity of the ecological resilience subsystem, the capacity of the resource environment subsystem, and the development capacity of the socio-economic subsystem. The spatial pattern of ecological carrying capacity of the Dongting Lake area from southwest to northeast is generally a “W” shape (higher–lower), with significant county differences in the inverted “U” shape. Shen et al. [7] used the state–space method to construct the ecological carrying capacity index system from three aspects, social coordination, resource and environmental support, and ecological resilience, and analyzed the spatial and temporal patterns of the middle reaches of the Yangtze River urban agglomeration, and the overall ecological carrying capacity index showed a fluctuating, upward “W” type trend. Zhang and Shi [8] used the ecological footprint model to construct the ecological carrying capacity system by selecting biological resources and energy resources data, and the results showed the ecological footprint per capita of the middle reaches of the Yangtze River urban agglomeration as a whole and Hubei, Hunan, and Jiangxi provinces showed “inverted U-shaped” characteristics. Feng et al. [9] used the biological abundance index and the ecological potential index of arable land to construct an evaluation system from three perspectives: the support capacity of the ecological resilience subsystem, the capacity of the resource and environment subsystem, and the development pressure of the socio-economic subsystem. Zhang et al. [10] used the pressure–state–potential (PSP) conceptual model with the help of GIS technology to construct the index systems of ecological pressure, ecological carrying state, and ecological carrying potential and obtained the conclusion that the spatial pattern of ecological carrying capacity suitability of sub-elements in the coastal region of Jiangsu showed various trends, and the same index had significant differences in magnitude, area, proportion, and layout in different cities and counties. Li et al. [11] used the pressure–state–response (PSR) model to select indicators in terms of forest support, social pressure, and system response, and the results showed that the overall forest ecological carrying capacity condition in Anhui Province has been greatly improved and most of the districts and counties in the province are at the loadable level.

Another aspect is the study of the coupling relationship between new urbanization and the ecological environment. The idea of coordination between urbanization, resources, and the environment can be traced back to the Socratic era when people began to pay attention to the relationship between people and the environment [12]. The early urban planning ideas in the West incorporated natural environmental factors into the overall planning of towns and cities, which is considered the germ of the coordination between urbanization, resources, and the environment [13]. Since modern times, the locational theory represented by Hettner and Hartshorne has been gradually introduced, and the idea of promoting the harmonious development of towns, resources, and the environment has been further developed [14]. Subsequently, Howard’s idyllic city theory was born, and the study of green urbanization entered a new stage, the core of which is still the harmonious development of towns and the environment [15]. Inspired by the idyllic city theory, the ecological concept reflected in the urbanization process has gradually developed toward systematization and standardization. Between the 1960s and 1970s, foreign scholars used various methods and models to study the relationship between urbanization and the ecological environment empirically. The existence of the “environmental Kuznets inverted U-shaped curve (EKC)” has become a research component of coordinated environmental and economic development [16,17,18]. In an in-depth study of the correlation between environment and urban development, some scholars [19,20,21,22,23,24] have used system dynamics (SD) models and Granger causality tests, as well as comprehensive evaluation models. In terms of the urban evaluation system, most scholars mainly construct urban evaluation index systems from demographic, economic, social, and spatial aspects and establish a pressure–state–response (PSR) model of resources and environment. In addition, some scholars [25,26] have integrated the concept of green objectives of new urbanization into resources and the environment and established a system model for the sustainable and coordinated development of population, resources, ecology, and the environment. Bi et al. [27] constructed an environment–economy coordination coefficient model based on the sustainable development system. In addition, some scholars [28] have established a model of the level of coupled collaborative development under a complex economic and environmental system based on mathematical operations. Yang [29] used the coupled coordinated development model of towns and ecosystems to classify the coordinated development level based on ecological theory. Some other scholars [30,31] have used the coordinated development model to study the coupling of economic urbanization and the environment in coastal areas. Most other studies [32,33,34] have explored the level of coordinated development between urbanization, resources, and the environment from the perspective of cities. In recent years, scholars have started to analyze the non-coordinated coupling degree between new urbanization and the ecological environment from the perspective of reverse thinking. At present, there have been fewer studies on the non-coordinated coupling between systems, and the so-called non-coordinated coupling refers to the development mechanism of urbanization and ecological environment to produce non-benign interactions. Spatio-temporal factors function with a coupling relationship between new urbanization and ecological environment, containing a variety of coordinated and non-coordinated coupling states, so some scholars analyze the new urbanization and ecological environment system based on reverse thinking. Furthermore, the problem can be diagnosed by analyzing non-coordinated pathology, which is of guiding significance for new urbanization and ecological environment leading to coordinated development. Sun [35] analyzed the coordinated development of urbanization and ecological environment in Jiangsu Province from the perspective of analytical pathology by using the non-coordinated coupling model thinking and obtained the conclusion that the non-coordinated coupling degree of urbanization and ecological environment in Jiangsu Province is continuously decreasing but always stays in the stage of grinding non-coordinated coupling. By analyzing the non-coordinated coupling relationship between urbanization and ecological environment in Shandong Province, Ren [36] combined spatial analysis methods to diagnose the influencing factors of their non-coordinated coupling characteristics and obtained the conclusion that the level of urbanization, ecological environment, and their non-coordinated coupling degree in Shandong Province showed a spatial pattern of superiority of the Ludong and Luzhong regions over western regions. Liang and Chen [37] argued that the development mechanism of urbanization and ecological environment produces non-benign interaction, which is a problem of regional economic development with a certain inevitability.

Synthesizing previous studies, we find that there is room for further expansion of research. First, the index system of new urbanization and ecological environment in the Yellow River Basin is intricate, but the existing index system cannot completely reflect the connotation of new urbanization and the complexity of the ecological environment. Most scholars construct the new urbanization index system from the perspectives of population, economy, and space. The development status of the Yellow River Basin now is different from that in the past, and it is impossible to quantify new urbanization with people as the core and to take into account high-quality development with past methods. Scholars have used different ways to quantify ecological carrying capacities, such as the comprehensive evaluation method, the ecological footprint method, the state–space method, and the pressure–state–response (PSR) model. However, the ecological environment itself has complex and diverse characteristics. For example, the evaluation of water resources by comprehensive evaluation only considers surface water resources but not groundwater resources; the ecological footprint method measures the consumption of resources by paying attention to direct consumption but not indirect consumption, ignoring other influencing factors in resource utilization. This work argues that using only a single method cannot show the complex situation of the regional ecological environment in detail. This paper adds the subsystems of the public management level and the science and technology input level to the index system of new urbanization based on previous research. The public management level reflects the perfect degree of the social management mechanism, and the science and technology input shows the development level of regional high-tech industry. The social management mechanism determines the lower limit of the happiness of residents’ lives, while the science and technology input determines the upper limit of human living standards. Therefore, adding the subsystems of the public management level and the science and technology input levels to the index system can more accurately measure the high-quality development level of new urbanization. For the ecological environment, this work selects indicators from the concept of ecological carrying capacity and five perspectives: resource utilization, energy consumption, government regulation and control, ecological pressure, and ecological elasticity.

In this paper, the resource utilization and government regulation subsystems are added based on a previous study. This paper measures the water resources carrying capacity indicators by referring to the water resources carrying capacity measurement method based on the water footprint theory by Huang et al. [38] and measures the spatial bearing capacity index of construction by referring to the measurement method in the “Territorial Resources and Environmental Carrying Capacity Evaluation Serving the Preparation of Territorial Planning” issued by the Ministry of Land and Resources. The per capita sown area of food crops represents agricultural land resources, and the per capita local financial expenditure on agriculture, forestry, and water affairs index represents the restoration power of soil and water resources, which not only quantifies urban soil and water resources in a more detailed way but also adds the dynamic influence of the human restoration of soil and water resources, and can measure the resource utilization in the process of new urbanization more accurately. In the new urbanization process, ecological and environmental protection engineering plays an important role, so this paper adds the government regulation subsystem to represent the execution power of government ecological and environmental engineering. Second, there are few domestic and foreign studies on the non-coordinated coupling of new urbanization and ecological environment, and domestic studies on the non-coordinated coupling of new urbanization and ecological carrying capacity have not yet appeared. Moreover, most of the existing domestic research of non-coordinated coupling is calculated based on the coupled coordination model. The results of the coupled coordination model mainly consider the horizontal spatial differences in the research object and do not take into account the vertical influence of the research object in time. This paper refers to the method of Wang et al. [39]. It improves the non-coordinated coupling model by adding time adjustment coefficients to the operation so that the model joins the adjustment of the interaction effect in time, which takes into account the system’s internal dynamics in time effects and brings the results closer to the actual situation.

## 2. Indicator System Construction and Method Selection

### 2.1. Construction of Indicator System

#### 2.1.1. New Urbanization

New urbanization is a comprehensive system of coordinated development of multiple elements. In order to reflect the requirements of people-oriented and high-quality development, this paper adds the index system of public management level and technology input level based on the traditional index system of new urbanization, in which the public management level represents the popularity of education, culture, medical care, and communication, which are the embodiments of the soft power of the city. The improvement of science and technology can improve residents’ living standards, which is a necessary condition for the construction of modern cities. Therefore, adding the public management level and the science and technology input level can reflect the new urbanization requirements of people-oriented and high-quality development. The new urbanization construction index system is shown in Table 1.

#### 2.1.2. Ecological Carrying Capacity

Traditional ecological carrying capacity usually revolves around ecological pressure, ecological elasticity, system response, and other aspects of selecting indicators for a comprehensive evaluation. However, the traditional method ignores the influence of potential resources and human regulation in measurement. For example, the traditional way of evaluating water resources is mainly for surface water, while other potential water resources, such as groundwater, have not been taken into account. As a reserve resource, groundwater resources are seldom paid attention to, but, as a part of carrying capacity, groundwater resources should also be considered. The evaluation method of land resource carrying capacity is mainly the static evaluation method; however, urban land carrying capacity should be dynamic. As the government manages some wastelands, the urban land carrying capacity will increase; in the process of resource consumption, urban land carrying capacity will also be reduced. Therefore, this paper selects indicators from five perspectives: resource utilization, energy consumption, government regulation, ecological pressure, and ecological elasticity. In the resource utilization subsystem, water and soil resources are measured by the water footprint method to measure the carrying capacity of water resources [38]. The urban construction space measurement method is used to measure the urban construction space carrying capacity, the per capita sown area of food crops represents the agricultural land carrying capacity, and the per capita local financial expenditure on agriculture, forestry, and water affairs represents the restoration capacity of urban water and soil resources, through which the full utilization of resources can be more accurately reflected. Nature can repair itself, but, in the process of new urbanization construction, the process of human-assisted repair is very important, so adding the government regulation subsystem represents the executive power of government ecological and environmental engineering. Based on these improvements, the traditional ecological carrying capacity system is combined with the entropy value method for objective assignment. The ecological carrying capacity index system is shown in Table 2.

### 2.2. Data Sources and Processing

#### 2.2.1. Data Sources

For this paper, we selected the data of the provinces along the Yellow River Basin from 2010 to 2017. As the State Council attributes Sichuan to the Yangtze River Economic Belt in the Yangtze River Economic Belt Development Plan, we excluded Sichuan Province from the study. The eight provinces (autonomous regions) of Qinghai, Gansu, Ningxia, Shaanxi, Shanxi, Henan, Inner Mongolia, and Shandong were taken as the research objects. The data in this paper were obtained from the China Statistical Yearbook, China Environmental Statistical Yearbook, provincial statistical yearbooks, and the Economy Prediction System (EPS) Global Statistical Platform. As the data of “three wastes” have not been updated since 2018, the data before 2018 were chosen for the study. In addition, the data involved in the water resources carrying capacity measurement in this paper were obtained from the water resources bulletin of each province, and the equilibrium factor was selected from the research results of Huang et al. [38] with the value of 5.19. The average production capacity of water resources was taken as 31.4 × 10^4^ m^3^/km^2^, and the regional water resources production factor was taken as the ratio of the regional water production modulus to the global multi-year average water production modulus [40]. Based on the setting of the Urban Land Classification and Planning and Construction Land Standard (GB50137-2011), the per capita land use standard was set after the hierarchical division of cities in the eight provinces of the Yellow River Basin based on reference to Sun [40] and the Notice on Adjusting the Standard of Urban Scale Classification issued by the State Council. Other indicators with individual missing data were filled in using the mean value method.

#### 2.2.2. Data Processing

Water resources carrying capacity

This paper draws on the idea of Huang et al. [38] and uses the water footprint theory to measure the carrying capacity of water resources. The water footprint theory can quantitatively analyze the relationship between human production activities and water resources. The theory is based on ecological footprint theory, which is easy to understand and can reflect the complete function of water resources, including virtual water resources, and the current academic research mostly adopts such methods. The calculation formulae are shown as follows:(1)$$W_{C}=P/AP=V_{T}/\left(S\times AP\right)$$
(2)$$Y_{W}=P/P_{g}$$
(3)$$C_{w}=N\times ac_{w}=0.4\times W_{C}\times Y_{w}\times (V_{T}/P_{g})$$
where $C_{w}$ represents the water carrying capacity; N represents the population; $ac_{w}$ is the per capita water carrying capacity; $W_{C}$ is the global average ecological productivity corresponding to the water equilibrium factor; $Y_{w}$ represents the water production silver; P is the regional water production modulus and $P_{g}$ is the global water production modulus, both representing the regional and global average water production capacity, respectively; $V_{T}$ represents the total regional water resources; S refers to the land area; and AP represents the global average ecological productivity of all organisms in a certain production area. The value 0.4 is the coefficient after deducting the production function to maintain the ecological environment.

2.Spatial carrying capacity of urban construction

This is calculated based on the idea of the “Evaluation of the environmental carrying capacity of national land resources serving the preparation of national land planning” issued by the Ministry of Land and Resources; the formula is as shown in (4):(4)$$C_{s}=S_{r}/S_{u}=\frac{S_{r}}{P_{a}\times A_{u}}$$
where $C_{s}$ represents the carrying capacity of urban construction space, $S_{r}$ is the actual space of the town, $S_{u}$ is the reasonable space size in line with sustainable development, $P_{a}$ is the total number of people, and $A_{u}$ is the standard of land use per capita. $C_{s}<1$ represents the redundant space of urban construction, $C_{s}=1$ represents that the town construction space is at the edge of full overflow, and $C_{s}>1$ represents that the carrying capacity has been unable to meet the development demand of new urbanization; that is, the actual size of urban construction space exceeds the size of reasonable space.

3.New urbanization and ecological carrying capacity measurement

For the index system of new urbanization and ecological carrying capacity, this paper adopts the extreme value method to eliminate the dimension and entropy method to carry out objective weighting for comprehensive evaluation.

When the indicators are positive indicators:(5)$$C_{s}=S_{r}/S_{u}=\frac{S_{r}}{P_{a}\times A_{u}}.$$

When the indicator is negative:(6)$$y_{ij}=\frac{\max(x_{j})-x_{ij}}{\max(x_{j})-\min(x_{j})}$$
where $y_{ij}$ denotes the value of the *j*th index in the *i*th sample after the extreme value method, $x_{ij}$ is the value of the *j*th index in the *i*th sample, $\min(x_{j})$ is the minimum value of the *j*th index in all samples in the total sample, and $\max(x_{j})$ is the maximum value of the *j*th index in the total sample. The weights of the new urbanization and ecological carrying capacity indicators are calculated using the entropy value method, as shown in Table 3.

### 2.3. Methodology Introduction

This paper uses an improved non-coordinated model to measure the non-coordinated coupling degree between new urbanization and the ecological carrying capacity system; the non-coordinated coupling model originates from the coupled coordination model of physics, which is usually used to analyze the interaction relationship between systems, and the non-coordinated coupling model is a more novel analysis model based on the reverse thinking perspective.

This paper draws on the non-coordinated coupling method proposed by Sun [35] and the time adjustment coefficient measure proposed by Wang et al. [39]. We propose introducing the non-coordinated coupling model with a time adjustment coefficient, i.e., the improved non-coordinated coupling model, and the basic principles are as follows:(7)$$C=2{ [\frac{U_{1}\times U_{2}}{{\left(U_{1}+U_{2}\right)}^{2}}]}^{\frac{1}{2}}$$
(8)$$U_{1}=\sum_{i=1}^{m}\alpha_{i}\times x_{i}$$
(9)$$U_{2}=\sum_{j=1}^{n}\beta_{j}\times y_{j}$$
(10)$$T=aU_{1}+bU_{2}$$
(11)$$D=\sqrt{C\times T}$$
where C represents the coupling degree of the two systems of new urbanization and ecological carrying capacity; for $C\in [0,1]$, when C=0, the coupling degree is the worst, which means there is no connection between the two systems and the development is also disorderly. *U*_1_, *U*_2_ represent the comprehensive development index of the two systems, $\alpha_{i}$, $\beta_{j}$ refer to the weight of each index, $x_{i}$, $y_{j}$ represent the dimensionless value of the index, and *m*, *n* represent the number of indexes. T represents the coupling degree between the systems. D is the coupling coordination degree between the systems, which reflects the coordination level between the new urbanization level and the ecological capacity. *a*, *b* are the coefficients to be determined; as China clearly proposes to develop the ecological protection and high-quality economic development of the Yellow River Basin simultaneously, the value of both is set to 0.5. The larger the value of the coordination degree, the higher the synergistic development between the systems. The uncoordinated coupling degree is further obtained on the basis of the coupling coordination degree as follows:(12)ND=1−D.

Referring to the approach of Wang et al. [39], the uncoupled coordination model is improved as follows:(13)$$r_{ik}=\frac{\left(\bar{X_{ik}}-\bar{X_{i(k-1)}}\right)}{\bar{X_{i(k-1)}}},R_{1}=\frac{\sum_{k=k_{\min}}^{k_{\max}}\sum_{i=1}^{m}\alpha_{i}\times r_{ik}}{k}$$
(14)$$r_{jk}=\frac{\left(\bar{X_{jk}}-\bar{X_{j(k-1)}}\right)}{\bar{X_{j(k-1)}}},R_{2}=\frac{\sum_{k=k_{\min}}^{k_{\max}}\sum_{j=1}^{n}\beta_{j}\times r_{jk}}{k}$$
(15)$${U^{\prime }}_{hk}=U_{hk}\times {\left(1-R_{h}\right)}^{k_{\max}-k}$$
where $\bar{X_{ik}}$ and $\bar{X_{jk}}$ are the mean values of the indicators of the new urbanization and ecological carrying capacity systems in the Yellow River Basin among the eight provinces, respectively, with i representing the new urbanization system, j representing the ecological carrying capacity system, and k representing the year; $r_{ik}$ and $r_{jk}$ are the growth rates of the indicators in the new urbanization and ecological carrying capacity systems relative to the previous year, respectively; $R_{1}$ and $R_{2}$ are the development adjustment coefficients of the scores of the new urbanization and ecological carrying capacity systems, respectively; $U_{hk}$ is the score of the hth system in the kth year, h=1,2, and ${U^{\prime }}_{hk}$ is the score of the new system adjusted by the development coefficient.

The obtained adjustment coefficients are substituted into Equations (10)–(12), and $ND^{\prime }$ is obtained as the value of the improved non-coordinated coupling degree. As there are few references for the non-coordinated classification, considering the existing literature on the division of non-coordinated coupling and improved coupling coordination [36,37,38,39], this paper divides the improved non-coordinated coupling level into six stages, as is shown in Table 4.

Among them, mild non-coordinated and low non-coordinated coupling states refer to the interaction between systems that develop benignly. The antagonistic non-coordinated coupling state refers to the inhibitory effect of two systems, but this state is still in the scope of benign interaction. The abrasive non-coordinated coupling state refers to when the development of coordination between the two systems from the original uncoordinated state is developing in the direction of coordination and benign, but, at present, is still in the pathological development state. Both high-level non-coordinated coupling and heavy non-coordinated coupling states indicate that the development between the systems is completely dysfunctional and the negative impact between the systems is large.

## 3. Results and Analysis

According to the new urbanization and ecological carrying capacity measurement, further measurement was carried out using the improved non-coordinated coupling model. The non-coordinated coupling degree values of new urbanization and ecological carrying capacity of the Yellow River Basin were obtained, as shown in Table 5.

### 3.1. Time-Series Feature Analysis

In order to make the radar map presentation clear, we drew a radar map of the non-coordinated coupling degree of each province in the Yellow River Basin from 2010 to 2017 with two years as one period. This was used to analyze the time-series characteristics and influence mechanism of the non-coordinated coupling degree of each province in the Yellow River Basin in combination with the stage division of non-coordinated coupling, as shown in Figure 1.

As can be seen from Figure 1, the values of non-coordinated coupling between new urbanization and ecological carrying capacity in the Yellow River Basin from 2010 to 2017 generally decrease, with an overall decrease of 52.04% and a large change. The study here is similar to the findings of Liu et al. [41], He et al. [42], Yang and Zhang [43], and Zhang and Shi [44]. In 2010, Shanxi Province, Henan Province, Qinghai Province, Gansu Province, and the Ningxia Hui Autonomous Region were all abrasive non-coordinated coupling. The rest of the provinces were in the state of antagonistic non-coordinated coupling. In 2017, the states of all provinces (autonomous regions) have improved, the Inner Mongolia Autonomous Region was at a low level of the non-coordinated coupling state, Shandong Province was a mild non-coordinated coupling state, and the rest of the provinces (autonomous regions) were in the antagonistic non-coordinated coupling state, in which the value of non-coordinated coupling in Shandong Province decreased from 0.315 to 0.015, the largest decrease (95.24%). Shaanxi Province decreased by about 41.77%, which was the smallest decrease. The conclusion here is similar to the study by Zhang and Shi [44] and slightly different from the study by Yang and Zhang [43], which might be mainly due to the fact that the rationalization and advanced industrial structure is considered in the economic development level in the study by Yang and Zhang [43].

In order to analyze the inner mechanism of non-coordinated coupling pathology, two perspectives of new urbanization and ecological carrying capacity were judged. From the weights of new urbanization and ecological carrying capacity indicators in Table 3, it can be seen that the main indicators affecting new urbanization are the resident population number, the number of water resources per capita, and all indicators of the science and technology input subsystem, i.e., the full-time equivalent of RD personnel of industrial enterprises above the scale and industrial enterprises’ RD expenditure. The main indicators affecting the ecological carrying capacity are the carrying capacity of water resources, the green coverage rate of built-up areas, per capita smoke (powder) emissions, and per capita sulfur dioxide emissions. Overall, in terms of new urbanization, from 2010 to 2017, the overall resident population in the Yellow River Basin increased by 3.1% and the investment in science and technology increased by 179%. Nevertheless, the per capita water resources decreased by 3.38%. In terms of ecological carrying capacity, the overall water resources carrying capacity of the Yellow River Basin decreased by 22.73% and the green coverage of built-up areas decreased by 10.34%.

In comparison, the per capita soot emissions and sulfur dioxide emissions per capita decreased by 32.07% and 53.69%, respectively. The main factor affecting the coordinated development of new urbanization and ecological carrying capacity is water resources, and more investment or policy support should be provided for water resources matters. Locally, the per capita water resources in the Inner Mongolia Autonomous Region and Shaanxi Province show a significant decrease, with 22.12% in the Inner Mongolia Autonomous Region and 13.66% in Shaanxi Province. Water resources are the basis of all production and economic activities; therefore, the Inner Mongolia Autonomous Region and Shaanxi Province need to increase water conservancy construction to enhance water resources supply and promote the new urbanization process. The water resources carrying capacity of Shandong Province decreased from 128.22 in 2010 to 68.41 in 2017 (46.65%). The green coverage of built-up areas decreased from 41.47% to 38.92%, indicating the reduction in water resources carrying capacity in ecological and environmental protection construction in Shandong Province and the lack of attention to the greening of built-up areas. The full-time equivalent of R&D personnel in industrial enterprises above designated size in Qinghai Province decreased by 2.36%, and the greening rate in built-up areas decreased by 7.36%, indicating that Qinghai Province needs to increase investment in science and technology personnel in the process of new urbanization and should focus on solving the problem of the built-up area greening rate in the ecological, environmental protection project.

### 3.2. Spatial Feature Analysis

#### Spatial Clustering Analysis

The mean values of the non-coordinated coupling degree of the new urbanization and ecological carrying capacity of the Yellow River Basin in Table 5 were subjected to a cluster analysis, and the results are shown in Table 6.

As seen in Table 6, the clustering results of the non-coordinated coupling degree between new urbanization and ecological carrying capacity in the Yellow River Basin during 2010–2017 are divided into three types. Type 1 is low-level non-coordinated coupling; only Shandong Province is of this type. Type 2 is antagonistic non-coordinated coupling; most provinces (autonomous regions) show this state, specifically Shanxi Province, Henan Province, Shaanxi Province, the Inner Mongolia Autonomous Region, and the Ningxia Hui Autonomous Region. Type 3 is the abrasive uncoordinated coupling state; the Gansu Province and Qinghai Province are in this state. On the whole, the non-coordinated coupling state of new urbanization and ecological carrying capacity of the provinces (autonomous regions) in the Yellow River Basin are better in the lower reaches of the Yellow River than in the middle and upper reaches. This study is similar to the findings of Liu et al. [41] and He et al. [42].

From the perspective of new urbanization, new urbanization is centered on people. Analyzing the factors affecting the development differences in the three types of regions, the regional resident population number is the main influencing factor of the new urbanization process. The resident population number in the low-level non-coordinated coupling region increased more than the other two types of regions during 2010–2017, increasing 3.38%. The resident population in the antagonistic non-coordinated coupling area increased by 2.46%, and the resident population number in the abrasive non-coordinated coupling area increased by 2.56%. Analyzing the factors affecting regional development differences from ecological carrying capacity, the green coverage of built-up areas in all three types of areas decreased during 2010–2017. However, the green coverage of built-up areas in low-level non-coordinated coupled areas decreased by 2.55%, while the decrease in antagonistic non-coordinated coupled areas was 8.58% and the decrease in abrasive non-coordinated coupled areas was 18.13%. Therefore, to reduce the gap between regions, each region should start to develop corresponding solutions from the perspective of the resident population number and the greening rate of built-up areas.

Based on the evaluation results of the improved non-coordinated coupling model of new urbanization and ecological carrying capacity in the Yellow River Basin from 2010 to 2017, global autocorrelation and local autocorrelation analyses of the non-coordinated coupling values of new urbanization and ecological carrying capacity in the Yellow River Basin were conducted with the help of ArcGIS software. The Moran’s I indices based on the global autocorrelation analysis model to calculate the non-coordinated coupling degree of each province (autonomous region) in the Yellow River Basin from 2010 to 2017 are shown in Table 7.

As can be seen from Table 7, the Moran’s I index of the non-coordinated coupling degree of new urbanization and ecological carrying capacity of each province in the Yellow River Basin from 2010 to 2017 passed the p-value test with a significance of 5%, indicating that the spatial distribution of the non-coordinated coupling degree of each province (autonomous region) in the Yellow River Basin is not random and there is a significant spatial autocorrelation. Looking at the overall global autocorrelation coefficient change trend, the spatial dependence of the non-coordinated coupling degree of each province (autonomous region) in the Yellow River Basin shows a trend of weakening, then strengthening, and then weakening with the evolution of time. In contrast, the correlation between the non-coordinated coupling degree of each province (autonomous region) shows a weakening trend.

Using the average level of the non-coordinated coupling degree of new urbanization and ecological carrying capacity of each province in the Yellow River Basin from 2010 to 2017 for local autocorrelation analysis, the local correlation results of non-coordinated coupling degree of new urbanization and ecological carrying capacity in the Yellow River Basin are obtained at the 10% significance level as shown in Figure 2.

From the results of local correlation analysis in Figure 2, it can be seen that the non-coordinated coupling degree of each province in the Yellow River Basin during 2010–2017 has spatial clustering characteristics and large differences in spatial distribution, showing an overall trend of better eastern than western reaches, and better middle and lower reaches than upper reaches of the Yellow River. Among them, low–low agglomeration areas are mainly distributed in Shaanxi Province and Shandong Province in the middle and lower reaches of the Yellow River. From the type point of view, Shandong Province belongs to the low-level non-coordinated coupling state and Shaanxi Province belongs to the antagonistic non-coordinated state, indicating that the non-coordinated coupling degree of the two provinces is in a better state and has a radiation effect centered on the two provinces, which has a benign influence on the surrounding areas. The high–high agglomeration areas are mainly distributed in the upper reaches of the Yellow River, centered on Qinghai Province, Gansu Province, and the Ningxia Hui Autonomous Region. Regarding type, both Qinghai Province and Gansu Province belong to the abrasive non-coordinated coupling state and the Ningxia Hui Autonomous Region belongs to the antagonistic non-coordinated coupling state, indicating that these areas have a poor coordinated development state compared with other areas, with high values of non-coordinated coupling degree, and have a non-benign impact on the surrounding areas.

Analyzing the intrinsic mechanism influencing spatial agglomeration from the perspective of new urbanization, the resident population and scientific and technological investment in Shaanxi Province and Shandong Province are much higher than those in Gansu Province, Qinghai Province, and the Ningxia Hui Autonomous Region. These two provinces can generate a radiation influence effect on central cities and promote the development of new urbanization in the surrounding regions through population flow and scientific and technological exchanges. At the same time, Gansu Province, Qinghai Province, and the Ningxia Hui Autonomous Region need to enhance the attractiveness of their cities to introduce a foreign population and increase the investment in science and technology to promote the development of their new urbanization in order to reduce the negative impact. From the perspective of ecological carrying capacity, the per capita smoke (powder) emissions and per capita sulfur dioxide emissions in Shaanxi Province and Shandong Province decreased significantly between 2010 and 2017, with 33.76% and 65.07% in Shaanxi Province and 50.15% and 70.26% in Shandong Province, respectively, far exceeding those of Gansu Province and Qinghai Province. This indicates that Shaanxi Province and Shandong Province pay more attention to pollution emission regulation, which improves the carrying level of their ecological carrying capacity and produces a reasonable and coordinated development mechanism with the rapidly developing new urbanization; then the high-quality environment also has a positive impact on the surrounding areas and improves the carrying level of the ecological carrying capacity of the nearby areas. Although Gansu Province, Qinghai Province, and the Ningxia Hui Autonomous Region also have a certain degree of emission regulation, its strength is much less than that in Shaanxi Province and Shandong Province.

## 4. Policy Implications

The policy enlightenment of this paper lies in the following points:The coordinated development of the new urbanization and ecological carrying capacity in the Yellow River Basin needs the two systems to be improved simultaneously to achieve a reasonable development mechanism of “green water and green mountains, that is, Jinshan and Yinshan”. The decline in water resources also affects the process of new urbanization and the ecological carrying capacity. Therefore, several areas with this problem need to increase the investment in water conservancy projects and increase the amount of afforestation to ensure the reduction in water and soil loss.In terms of increasing the number of permanent residents, we should increase regional scientific and technological investment at the national level, regulate environmental pollution emissions, and promote the development of tertiary industry. On the basis of certain sustainable development, we should attract investment through preferential policies and the establishment of funds, so as to lead society and large, medium, and small enterprises to actively participate in market-oriented construction and to achieve the purpose of improving the attractiveness of the city and the happiness of the people.In terms of investment in science and technology, we should promote technical exchanges between the middle and lower reaches and the upper reaches of the Yellow River Basin; provide Counterpart Assistance between provinces, cities, and counties; strengthen scientific and technological cooperation among regional industrial chains; break the barriers of provincial administrative boundaries; and tap the scientific and technological development potential of the Jinzhong urban agglomeration, Hexi urban agglomeration, and other cities.In terms of improving the greening rate of urban built-up areas, we should put forward scientific and reasonable standards for the green coverage rate of construction areas for urban planning departments, real estate developers, and other relevant departments and industries and establish a strict supervision system to make new urbanization develop toward green and coordinated urbanization.

## 5. Conclusions

Taking eight provinces (autonomous regions) in the Yellow River Basin as the research object, in this paper we constructed a new urbanization and ecological carrying capacity evaluation index system. On this basis, the uncoordinated coupling level of new urbanization and ecological carrying capacity in the Yellow River Basin was measured by using the improved uncoordinated coupling model, and its temporal and spatial characteristics and internal impact mechanism were analyzed. We draw the following conclusions:The coordinated development of new urbanization and ecological carrying capacity in the Yellow River Basin generally shows a benign development trend, and the uncoordinated coupling state of all provinces decreased from a higher level to a lower level. Specifically, the uncoordinated coupling level of new urbanization and ecological carrying capacity in Shandong Province has achieved a great leap, from antagonistic uncoordinated coupling to mild uncoordinated coupling. In other regions, the uncoordinated coupling between the two is mainly from running-in to antagonizing.There is a spatial agglomeration effect between the new urbanization and the uncoupled level of ecological carrying capacity in the Yellow River Basin. The degree of uncoordinated coupling among regions in the Yellow River Basin can be divided into three categories. The first is the low-level uncoordinated coupling state, mainly in Shandong Province. The second category, antagonistic uncoordinated coupling state, mainly includes: Shanxi Province, the Inner Mongolia Autonomous Region, Henan Province, Shaanxi Province, and the Ningxia Hui Autonomous Region. The third is the running-in uncoordinated coupling state, which mainly includes Gansu Province and Qinghai Province.The uncoupled level of new urbanization and ecological carrying capacity in the Yellow River Basin has a spatial spillover effect. There are mainly two types of spatial spillover effects: low–low agglomeration and high–high agglomeration. Low–low concentration areas are mainly distributed in Shaanxi Province and Shandong Province in the middle and lower reaches of the Yellow River, while high–high concentration areas are mainly distributed in the upper reaches of the Yellow River, with Qinghai Province, Gansu Province, and the Ningxia Hui Autonomous Region as the center.The main factors that affect the uncoordinated coupling degree between new urbanization and ecological carrying capacity in the Yellow River Basin are water resources, population, scientific and technological investment, and the green coverage of built-up areas. Among them, water resources are not only the main factor restricting the new urbanization level of the Yellow River Basin but also the key factor affecting its ecological carrying capacity level. The main factors affecting the level of new urbanization among provinces are the number of people and the level of scientific and technological investment. The main factor affecting the level of ecological carrying capacity among provinces is the green coverage of built-up areas.

This paper mainly analyzed the non-coordination between the new urbanization and ecological carrying capacity in the Yellow River Basin. It provides some conclusions for reference, however, this paper does not analyze urban administrative area and below due to limited data, and the inner dynamic mechanism between coordinated coupling and non-coordinated coupling was not analyzed, which needs to be studied in depth in the future.

## Figures and Tables

**Figure 1 ijerph-19-09016-f001:**
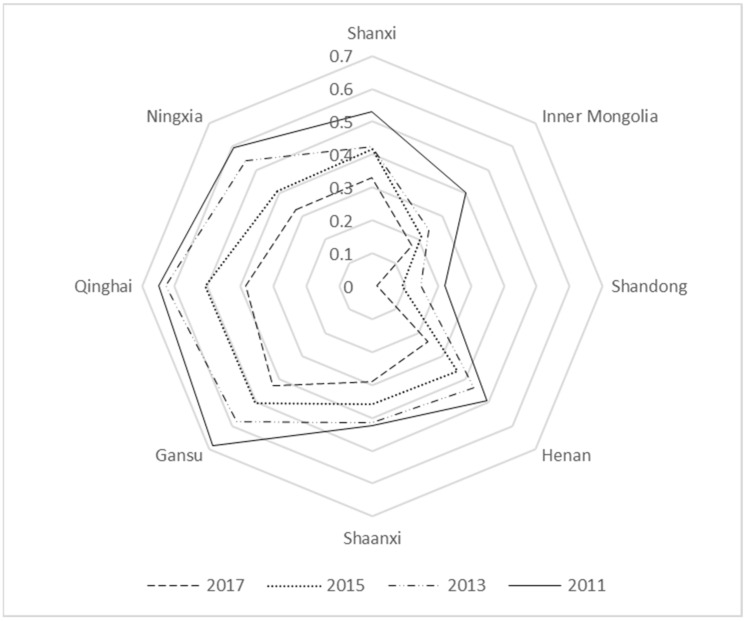
Radar chart of uncoordinated coupling degree between new-type urbanization and ecological carrying capacity in the Yellow River Basin.

**Figure 2 ijerph-19-09016-f002:**
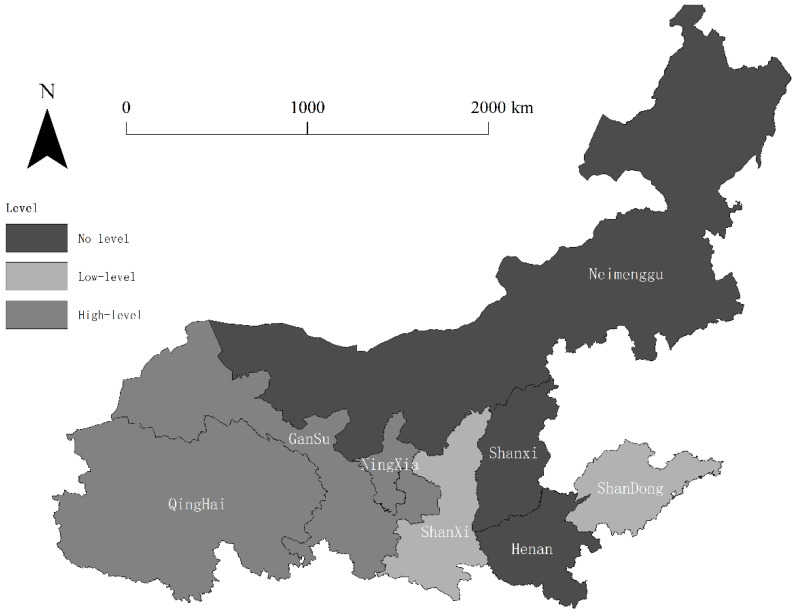
Local correlation diagram of uncoordinated coupling degree in the Yellow River Basin. Note: This map was made based on the standard map (GS(2020)4634) downloaded from the standard map service website of the National Administration of Surveying, Mapping, and Geographic Information. The base map has not been modified.

**Table 1 ijerph-19-09016-t001:** Indicator system of new urbanization.

Target Layer	Criteria Layer	Index Layer	Index Attribute	Index of the Unit
A new type of urbanization level	Population development level	Permanent population	+	People
		Population density	-	People/square kilometers
		Urbanization rate	+	%
	Level of economic development	Investment in fixed assets per capita (excluding rural households)	+	RMB/person
		Proportion of employees in secondary and tertiary industries	+	%
		Per capita local fiscal revenue	+	RMB/person
		Retail consumption per capita	+	RMB/person
		Disposable income of urban residents	+	RMB
		GDP per capita	+	RMB
		Ratio of added value of secondary and tertiary industries in GDP	+	%
	Level of public administration	Volume of books in the library per 100 people	+	Copies of books
		Number of hospital and health hospital beds per 1000 people	+	Beds
		Mobile phones per 10,000 people	+	set
		Teachers per 10,000 people	+	People
	Level of spatial development	Per capita road area	+	m^2^/person
		Water resources per capita	+	Cubic meters per person
		Per capita built-up area	+	m^2^/person
	Investment level of science and technology	RD personnel of industrial enterprises above designated size is equivalent to full-time equivalent	+	People/year
		RD expenditure of industrial enterprises above designated size	+	Ten thousand RMB

**Table 2 ijerph-19-09016-t002:** Index system of ecological carrying capacity.

Target Layer	Criteria Layer	Index Layer	Index Attribute	Index of the Unit
Ecological carrying capacity	Resource utilization	Water carrying capacity	+	-
		Spatial carrying capacity of urban construction	-	-
		Per capita local government spending on agriculture, forestry, and water conservancy	+	RMB/person
		Per capita sown area of grain crops	+	Hectare/person
	Energy consumption	Natural gas consumption per capita	-	Cubic meter/person
		Per capita coal consumption	-	Tons/person
		Per capita gasoline consumption	-	Tons/person
		Per capita electricity consumption	-	Ten thousand kilowatt hours per person
		Per capita water consumption	-	m^3^/person
	Government regulation	Per capita soil erosion control area	+	Hectare/person
		Completed investment of waste gas treatment project	+	Million RMB
		Completed investment of waste water treatment project	+	Million RMB
		Per capita local government spending on environmental protection	+	RMB/person
		Completed investment of industrial pollution control	+	Million RMB
		Area of artificial afforestation in the current year	+	Thousands of hectares
		Proportion of investment in environmental protection in GDP	+	%
		Per capita soil erosion control area	+	Hectare/person
		Personnel employed in urban units in the management of water conservancy, environment, and public facilities	+	Ten thousand people
	Ecological pressure	Per capita output of industrial solid waste	-	Million tons/person
		Per capita smoke (dust) emissions	-	Million tons/person
		Sulfur dioxide emissions per capita	-	Million tons/person
		Wastewater discharge per capita	-	Million tons/person
		Engel’s coefficient for urban residents	-	%
	Ecological elasticity	Sewage treatment rate	+	%
		Per capita green park area	+	Cubic meter/person
		Green coverage rate in built-up areas	+	%
		Household garbage harmless disposal rate	+	%
		Comprehensive utilization rate of industrial solid waste	+	%

**Table 3 ijerph-19-09016-t003:** New urbanization and ecological carrying capacity weight system.

Information	Indicators	Weight
New urbanization	Permanent population	0.079
	Population density	0.030
	Urbanization rate	0.019
	Investment in fixed assets per capita (excluding rural households)	0.030
	Proportion of employees in secondary and tertiary industries	0.024
	Per capita local fiscal revenue	0.028
	Retail consumption per capita	0.038
	Disposable income of urban residents	0.026
	GDP per capita	0.027
	Ratio of added value of secondary and tertiary industries in GDP	0.020
	Volume of books in the library per 100 people	0.034
	Number of hospital and health hospital beds per 1000 people	0.017
	Mobile phones per 10,000 people	0.014
	Teachers per 10,000 people	0.036
	Per capita road area	0.055
	Water resources per capita	0.199
	Per capita built-up area	0.058
	RD personnel of industrial enterprises above designated size is equivalent to full-time equivalent	0.116
	RD expenditure of industrial enterprises above designated size	0.150
Ecological carrying capacity	Water carrying capacity	0.075
	Spatial carrying capacity of urban construction	0.043
	Per capita local government spending on agriculture, forestry, and water conservancy	0.015
	Per capita sown area of grain crops	0.032
	Natural gas consumption per capita	0.027
	Per capita coal consumption	0.012
	Per capita gasoline consumption	0.028
	Per capita electricity consumption	0.025
	Per capita water consumption	0.027
	Per capita soil erosion control area	0.022
	Completed investment of waste gas treatment project	0.018
	Completed investment of waste water treatment project	0.022
	Per capita local government spending on environmental protection	0.021
	Personnel employed in urban units in the management of water conservancy, environment, and public facilities	0.020
	Completed investment of industrial pollution control	0.015
	Area of artificial afforestation in the current year	0.026
	Proportion of investment in environmental protection in GDP	0.017
	Per capita soil erosion control area	0.016
	Per capita output of industrial solid waste	0.059
	Per capita smoke (dust) emissions	0.079
	Sulfur dioxide emissions per capita	0.075
	Wastewater discharge per capita	0.059
	Engel’s coefficient for urban residents	0.052
	Sewage treatment rate	0.058
	Per capita green park area	0.049
	Green coverage rate in built-up areas	0.060
	Household garbage harmless disposal rate	0.033
	Comprehensive utilization rate of industrial solid waste	0.015

**Table 4 ijerph-19-09016-t004:** Phase division of uncoordinated coupling.

Non-Coordinated Coupling Degree (ND)	Type	Characteristics
(0.0, 0.1)	Mild non-coordinated coupling	Development between systems tends to be benign interaction, and the positive influence is obvious.
(0.1, 0.2)	Low non-coordinated coupling	
(0.2, 0.5)	Antagonistic non-coordinated coupling	There is a partial imbalance between the systems, and the overall development is relatively complex.
(0.5, 0.8)	Abrasive non-coordinated coupling	
(0.8, 0.9)	High non-coordinated coupling	System is completely out of balance, the development is unbalanced, and the negative impact is obvious.
(0.9, 1.0)	Heavy non-coordinated coupling	

**Table 5 ijerph-19-09016-t005:** Values of uncoordinated coupling degree.

Administrative Areas	2010	2011	2012	2013	2014	2015	2016	2017
Shanxi	0.576	0.531	0.483	0.424	0.425	0.415	0.362	0.331
Inner Mongolia	0.479	0.402	0.354	0.242	0.227	0.213	0.189	0.172
Shandong	0.315	0.221	0.187	0.147	0.082	0.092	0.035	0.015
Henan	0.513	0.494	0.469	0.435	0.367	0.364	0.291	0.239
Shaanxi	0.498	0.423	0.415	0.416	0.389	0.360	0.339	0.290
Gansu	0.781	0.687	0.611	0.583	0.545	0.504	0.454	0.426
Qinghai	0.786	0.647	0.527	0.626	0.552	0.506	0.451	0.384
Ningxia	0.610	0.595	0.521	0.541	0.440	0.409	0.335	0.329

**Table 6 ijerph-19-09016-t006:** Spatial clustering results.

Provinces	Type
Shandong	Low non-coordinated coupling
Shanxi, Inner Mongolia, Henan, Shaanxi, Ningxia	Antagonistic non-coordinated coupling
Gansu, Qinghai	Abrasive non-coordinated coupling

**Table 7 ijerph-19-09016-t007:** Global autocorrelation Moran’s I index.

Indicators	2010	2011	2012	2013	2014	2015	2016	2017
Moran’s I index	0.279	0.237	0.149	0.249	0.186	0.160	0.134	0.155
P	0.002	0.006	0.015	0.002	0.008	0.021	0.023	0.015
Z	3.625	3.329	2.77	3.403	3.006	2.772	2.633	2.789

## Data Availability

The data in this paper are obtained from China Statistical Yearbook, China Environmental Statistical Yearbook, provincial statistical yearbooks, water resources bulletins and EPS Global Statistical Data Platform, the Urban Land Classification and Planning and Construction Land Standard (GB50137-2011) issued by the Bureau of Land and Resources, and the Notice on Adjustment of Urban Scale Classification Standard issued by the State Council.

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
