# Peer review of "Uncoordinated Coupling Assessment of New Urbanization and Ecological Carrying Capacity in the Yellow River Basin"

_ijerph, 2022, doi:10.3390/ijerph19159016_

Round 1
Reviewer 1 Report
Abstract is way too long, keep it within word limit. Supplement literature sources.
A few tables have inconsistent capitalization (like “information” and “The weight” in Table 3 for example). Consider picking one and sticking to it.
There are a lot of run-on sentences that could be broken down into smaller ones.
Lines 24, 31, 42 and 45: After a semicolon the sentence continues from a lower-case letter, did you mean to use a full stop?
Line 96: The sentence is very long and does not have a full stop at the end either. Fully finish the end of the sentence and if possible, separate the sentence into several smaller ones.
Start of line 105: Another long sentence that can be broken down into smaller ones.
Line 142: Selected biological what? Sentence seems unfinished or maybe cut off too early since the next sentence seems to be a continuation.
Line 144: It should be “used such a biological”.
Lines 199-200: It should be “Furthermore, the problem can be diagnosed by analyzing …”
Line 202: It should be “coupling model with a thought to analyze …”
Lines 227-229: “This paper” is used twice in a row to start sentences. Consider using something like “This work” instead of one of those times.
Lines 229-230: It should be “add the subsystems of public management and science and technology input levels for the index system …”
Table 1: It is hard to tell apart which index layer belongs to which criteria level, maybe separate the categories in a more obvious way like in Table 2.
Line 290: It should be “capacity should be a dynamic change”.
Lines 290-292: The sentence should not start with “because” (maybe use “since” instead) and the whole sentence itself feels like an unfinished fragment, consider revising.
Line 322: The capacity is missing superscript, it should be “m3/km2” instead.
Formulas 12 and 15: It seems that these formulas have glitched in the pdf version, check to see if there is a way to fix this or rewrite them.
Table 4: “Characteristics of the” what? It seems unfinished.
Figure 1: It seems the figure messes up line counting, see if the correct alignment is used.
Reviewer 2 Report
Congratulations for the job.
Some points:
1. Title: New urbanization or urbanization (alone) or recent urbanization? Think about it because for the reader it is confusse.
2. The abstract is too long, maybe it could be resumed.
3. The introduction is good, but cite much of studies, but not give the result of none. Maybe, due to the objetive of the manuscript is improve the index of valoration, could be interesting to know the result (at least of the principal ones or with the studies has more similar characteris).
4. About the data, specially water resources, the index could be improved taking into account regenerated water or dessalation water posibilities. It is also important take into account the water squeme, principal uses (agriculture, industry, population, etc. - demands), climate change scenario (not analysed). Ground water usually are analysed like a reserve of water for scarcity moments, not to use as other more resource source (it is important to analyse this point too and corroborate this perspective with the water management politics).
Other less important points:
1. Abbreviation: it is important for the reader that the first time its appears an abbreviation, it is explained (line 149 (PRS), line 172 (EKC), line 316 (EPS), for example).
2. Citation homogeneity. During the manuscript it is use two ways to cite a study
Author/s - [number] - argument.
Author/s - argument - [number].
3. Table 1 and table 4: Horizontally delimit the table by row. It is difficult to undestand.
4. Table 3: the sum of the weigths is not one, in the first case is almost one, but in the second case (ecological carrying capacity) is 1,016. Check and correct if it is necessary.
5. Table 6. Capital letter for "S"handong
About the conclusions congratulations, are simple and interesting.
